# Recommendations to Improve Tick-Borne Encephalitis Surveillance and Vaccine Uptake in Europe

**DOI:** 10.3390/microorganisms10071283

**Published:** 2022-06-24

**Authors:** Michael Kunze, Pavle Banović, Petra Bogovič, Violeta Briciu, Rok Čivljak, Gerhard Dobler, Adriana Hristea, Jana Kerlik, Suvi Kuivanen, Jan Kynčl, Anne-Mette Lebech, Lars Lindquist, Iwona Paradowska-Stankiewicz, Srđan Roglić, Dita Smíšková, Franc Strle, Olli Vapalahti, Nenad Vranješ, Nataliya Vynograd, Joanna Maria Zajkowska, Andreas Pilz, Andreas Palmborg, Wilhelm Erber

**Affiliations:** 1Center for Public Health, Medical University of Vienna, 1090 Vienna, Austria; michael.kunze@meduniwien.ac.at; 2Ambulance for Lyme Borreliosis and Other Tick-Borne Diseases, Department of Prevention of Rabies and Other Infectious Diseases, Pasteur Institute Novi Sad, 21000 Novi Sad, Serbia; pavle.banovic@mf.uns.ac.rs; 3Department of Microbiology with Parasitology and Immunology, Faculty of Medicine in Novi Sad, University of Novi Sad, 21000 Novi Sad, Serbia; 4Department of Infectious Diseases, University Medical Centre Ljubljana, Japljeva 2, 1525 Ljubljana, Slovenia; petra.bogovic@kclj.si (P.B.); franc.strle@kclj.si (F.S.); 5Department of Infectious Diseases, “Iuliu Hațieganu” University of Medicine and Pharmacy Cluj-Napoca, 400348 Cluj-Napoca, Romania; briciu.tincuta@umfcluj.ro; 6University Hospital for Infectious Diseases “Dr. Fran Mihaljević”, Mirogojska 8, 10000 Zagreb, Croatia; rok.civljak@bfm.hr (R.Č.); sroglic@bfm.hr (S.R.); 7Department for Infectious Diseases, University of Zagreb School of Medicine, 10000 Zagreb, Croatia; 8National Reference Laboratory for TBEV, Bundeswehr Institute of Microbiology, 80937 Munich, Germany; gerharddobler@bundeswehr.org; 9Faculty of Medicine, Carol Davila University of Medicine and Pharmacy, 020022 Bucharest, Romania; adriana.hristea@umfcd.ro; 10Department of Epidemiology, Regional Authority of Public Health in Banská Bystrica, 97556 Banská Bystrica, Slovakia; kerlik@vzbb.sk; 11Department of Virology, Faculty of Medicine, University of Helsinki, 00014 Helsinki, Finland; suvi.kuivanen@helsinki.fi (S.K.); olli.vapalahti@helsinki.fi (O.V.); 12Department of Infectious Diseases Epidemiology, National Institute of Public Health, Vinohrady, 10000 Prague, Czech Republic; jan.kyncl@szu.cz; 13Department of Epidemiology and Biostatistics, Third Faculty of Medicine, Charles University, 10000 Prague, Czech Republic; 14Department of Infectious Diseases, Copenhagen University Hospital—Rigshospitalet, 2100 Copenhagen, Denmark; anne-mette.lebech@regionh.dk; 15Division of Infectious Diseases, Department of Medicine Huddinge, Karolinska Institute, 14186 Stockholm, Sweden; lars.lindquist@ki.se; 16Department of Epidemiology of Infectious Diseases and Surveillance, National Institute of Public Health, National Institute of Hygiene—National Research Institute, 00791 Warsaw, Poland; istankiewicz@pzh.gov.pl; 17Department of Infectious Diseases, Second Faculty of Medicine, Charles University, 18081 Prague, Czech Republic; dita.smiskova@bulovka.cz; 18Department of Veterinary Biosciences, University of Helsinki, 00014 Helsinki, Finland; 19Virology and Immunology, HUSLAB, Helsinki University Hospital, 00260 Helsinki, Finland; 20Department for Research & Monitoring of Rabies & Other Zoonoses, Pasteur Institute Novi Sad, 21000 Novi Sad, Serbia; nenad.vranjes@gmail.com; 21Department of Epidemiology, Danylo Halytsky Lviv National Medical University, 79010 Lviv, Ukraine; vynogradno@ukr.net; 22Department of Infectious Diseases and Neuroinfections, Medical University of Białystok, 15-540 Białystok, Poland; joanna.zajkowska@umb.edu.pl; 23Medical and Scientific Affairs, Pfizer Vaccines, 1210 Vienna, Austria; andreas.pilz@pfizer.com; 24Medical and Scientific Affairs, Pfizer Vaccines, 19138 Stockholm, Sweden; andreas.palmborg@pfizer.com

**Keywords:** tick-borne encephalitis (TBEV), Western Europe, incidence, Europe, surveillance, vaccine, vaccination, vaccine coverage, recommendations

## Abstract

There has been an increase in reported TBE cases in Europe since 2015, reaching a peak in some countries in 2020, highlighting the need for better management of TBE risk in Europe. TBE surveillance is currently limited, in part, due to varying diagnostic guidelines, access to testing, and awareness of TBE. Consequently, TBE prevalence is underestimated and vaccination recommendations inadequate. TBE vaccine uptake is unsatisfactory in many TBE-endemic European countries. This review summarizes the findings of a scientific workshop of experts to improve TBE surveillance and vaccine uptake in Europe. Strategies to improve TBE surveillance and vaccine uptake should focus on: aligning diagnostic criteria and testing across Europe; expanding current vaccine recommendations and reducing their complexity; and increasing public education of the potential risks posed by TBEV infection.

## 1. Introduction

Tick-borne encephalitis (TBE) is a disease of the central nervous system (CNS), caused by a virus in the *Flavivirus* genus that is predominantly transmitted to humans by tick bites, or in some cases, alimentary infection through the milk of animals infected with the TBE virus (TBEV) [1,2]. TBE is the most important tick-borne viral disease in Europe, endemic in 27 European countries [3,4,5], and represents a considerable disease burden.

There are at least five antigenic subtypes of TBEV. Among them, the European tick-borne encephalitis virus (TBEV-EU), the Siberian tick-borne encephalitis virus (TBEV-Sib), and the Far Eastern tick-borne encephalitis virus (TBEV-FE) are the main subtypes, of which TBEV-EU is predominant in Europe [6,7]. TBE may cause severe sequelae and quantifiable long-lasting limitations in daily life. In addition, postencephalitic syndrome, which can affect up to 50% of hospitalized TBE patients, can impact a patient’s quality of life for months to years after acute illness has resolved [8,9].

There has been an increase in reported TBE cases in the past two decades and TBE is spreading to new areas, including areas as high as 1000 m above sea level in Norway and 1500 m above sea level in central Europe, as well as regions previously believed to be free of the virus [10,11,12,13], highlighting the need for better management of TBE risk in Europe. Since 2012, the European Centre for Disease Prevention and Control (ECDC) requires all European Union (EU) member states, plus Iceland and Norway, to annually report their TBE data to the European Surveillance System (TESSy) [5]. Furthermore, TBE cases are likely underreported; only 8% of countries in Europe use the current ECDC diagnostic criteria, and while some national diagnostic criteria are similar to those of the ECDC, others are very different [14,15]. Other limitations also have an impact on the number of cases reported, including access to diagnostic testing and suboptimal surveillance systems, particularly regional and passive surveillance [10,11,16,17].

Vaccination remains the most effective method of protection against TBE [18]. In Europe, both available vaccines report seroconversion rates in a range of 92–100%, as measured by a commercial enzyme-linked immunosorbent assay (ELISA) or neutralization test [19,20,21,22,23]. High field vaccine effectiveness, >90%, has been demonstrated in several settings and across all ages, including people ≥60 years old [24,25]. However, breakthrough TBE, despite vaccination, has been noted across Europe [26,27]. Some authors quote that these vaccine failures appear to occur more often in patients >50 years of age, possibly due to age-related weakening of the immune system, though the exact cause has not been elucidated [27,28,29]. However, this is not generally accepted and recent studies do not confirm a higher vaccine failure rate in the elderly [23].

Owing to the varying endemicity of TBE across Europe, TBE vaccine recommendations vary considerably across Europe, including no specific TBE vaccine recommendations in some countries [10]. Uptake and compliance with TBE vaccination in Europe is also highly variable, with low overall vaccination rates [3].

In September and November 2021, two separate meetings with experts in TBE from 13 European countries were convened to discuss the effectiveness of European TBE surveillance systems, their impact on vaccine recommendations, and the role of vaccine awareness on vaccine uptake. As participants at these meetings, we present the discussion on these topics and provide suggestions to improve vaccine uptake in Europe.

## 2. Disease Epidemiology

In Europe, TBE is generally diagnosed in forested belts ranging from Norway, Southeastern UK, Eastern France, and the Netherlands, down to Northern Italy through Central and Eastern Europe (Figure 1) [10]. TBEV exists in natural foci, areas where TBEV is circulating among ticks and reservoir hosts. As such, TBE is restricted to particular geographical regions, leading to TBE endemic areas [30,31,32].

Indeed, the incidence of TBE is rising across Europe, including those at higher altitudes in Central Europe and further north and west in Scandinavian countries, as well as regions previously believed to be free of the virus [10,11,12,13]. There are wide fluctuations in the reporting of national annual cases, which is caused by many factors that will be described and discussed later in this section. However, it is of note that cumulative reported TBE cases across Europe in 2020 were twice those in 2015 (TBE case numbers: 2015 = 1779, 2020 = 3626), (Figure 2A) [10]. An increase in reported TBE case numbers was observed in 76% (13/17) of countries over this period (Figure 2B) [10]. Most notably, there was a more than three-fold increase from 2015 to 2020 in Germany, Italy, Norway, Slovenia, and Switzerland [10].

Aside from the Europe-wide rising incidence, there is also a shift in the pattern of national/regional risk areas. For example, TBE is endemic in Croatia. The total number of reported human TBE cases in Croatia from 2009 to 2020 ranged from 6 in 2016 (mean incidence of 0.15/100,000 inhabitants) to 45 in 2012 (mean incidence of 1.1/100,000 inhabitants) [33]. Historically, TBE cases were restricted to the northwestern continental parts of the country [34,35], and currently, endemicity remains highest in northwestern counties, with a mean incidence of 3.6–6.8/100,000 inhabitants [36]. However, the area of TBE circulation in Croatia is expanding with the emergence of several new natural foci [37].

The causes for this increase are unclear but are likely to include various factors, such as changing human behavior, improved diagnosis of TBE, and changes in tick habitats and periods of activity due to climate change [10]. The impact of national lockdowns during the COVID-19 pandemic in 2020 is unclear; however, TBE incidence across Europe did not differ significantly in 2020 compared with projected incidence values, suggesting that the COVID-19 pandemic may have had only a limited impact on the increased TBE cases observed in 2020 [38].

## 3. TBE Surveillance in Europe

The identification of TBE-endemic areas is essential to inform national and international programs of TBE risk management. Since 2012, the European Centre for Disease Prevention and Control (ECDC) requires all European Union (EU) member states, plus Iceland and Norway, to annually report their TBE data to the European Surveillance System (TESSy) [5]. However, comparison between regions and countries is dependent on the accuracy and consistency of national/regional surveillance systems.

### 3.1. TBE Case Definitions

Common TBE case definitions, officially introduced by the ECDC in 2012 and updated in 2018, aimed to increase and standardize TBE identification within surveillance programs across the EU [10,39,40]. However, TBE case definitions and reporting have been implemented inconsistently throughout Europe (Appendix A) [10,14]. The ECDC, in 2019, reported that 92% (23/25) of countries utilize an ECDC TBE case definition but that only 8% (2/25) utilize the current 2018 ECDC case definition [14]. In several countries that do not utilize the ECDC diagnostic criteria, such as Italy, national diagnostic criteria are largely similar to the ECDC criteria and, therefore, may not lead to major differences in reported TBE case numbers [41]. However, this may not be the case in countries that have key differences in requirements for the confirmation of a TBE case, such as in Germany, where there is no requirement for CNS symptoms [15,42,43]. Furthermore, TBE diagnostic criteria in non-EU member states, such as Ukraine and Serbia, often differ from ECDC diagnostic criteria [16,44,45]. Differences in case definitions between countries, therefore, make country data on TBE prevalence difficult to compare and result in varying degrees of accuracy [10].

TBEV infections often present as mild, with non-specific febrile illness, in the first phase of the disease, without impact on the CNS [46]. The ECDC case definition requires symptoms of inflammation in the CNS for confirmation of TBE, which occurs in the second phase of the disease only [10,39,46]. In several cases, the first phase of illness (fever only) does not progress to CNS inflammation, and in most countries, only cases following the ECDC criteria for inflammation of the CNS are reported. There are only a few countries—Austria, Latvia, Germany, and Slovenia—that collect data on nonspecific, non-CNS symptoms [15,24,47,48,49]. In addition, mild CNS symptoms may be missed and go un-reported since they would not fulfil the ECDC criteria, leading to underreporting of TBE. This is of particular note in pediatric patients, where symptoms are more often mild and can be misdiagnosed. This may lead to pediatric TBE cases being disproportionately underreported compared with adult cases [10,50], as shown by Hansson et al. [51], who demonstrated that up to two-thirds of TBE cases in children are missed.

### 3.2. Diagnostic Testing

ELISAs are the most common method of detecting antibodies against TBEV [32,52]. While ELISAs are available across Europe [32], education on when to test for TBEV infection and when other diagnostic tests are appropriate, and access to those tests (e.g., neutralization tests as well as PCR verification of genetic markers in attached ticks or the blood and cerebrospinal fluid of the patient), are essential for the accurate diagnosis of TBE.

TBE underreporting may be exacerbated by clinicians who do not test for TBEV infection because they do not recognize the possibility of CNS inflammation or, if they suspect CNS inflammation, may be less likely to perform a CSF examination supportive of a TBE diagnosis.

In addition, underreporting of TBE may occur in countries where TBEV and West Nile virus (WNV) infections are endemic, such as Serbia or Romania [16,53]. TBE cases may be misdiagnosed as West Nile encephalitis and vice versa because of antibody cross-reactivity in ELISA assays and the larger scale of WNV testing compared with TBEV testing in some countries [16,54]. Improved laboratory capacities and implementation of neutralization assays in these countries could improve TBE identification by distinguishing TBE from other flaviviral infections [16].

The choice of direct or indirect assay to detect TBEV is dependent on the phase of disease [52]. In the initial phase of illness, TBEV RNA can be detected in blood, whereas antibodies to TBEV are typically absent. In the second, meningo/encephalitic phase, IgM and IgG antibodies to TBEV are present but direct detection of viral RNA is typically unsuccessful. TBE following infection with TBEV-Sib and TBEV-FE is more commonly associated with CNS inflammation during a more rapid onset of a monophasic disease course. This may mean that patients infected with TBEV-Sib or TBEV-FE are more likely to be seronegative at time of hospitalization compared with TBEV-EU-infected patients and may lead to misdiagnosis of TBEV-Sib and TBEV-FE cases in countries where these strains are also present, such as Ukraine, Latvia, and Estonia [10].

### 3.3. Regional Surveillance

The tendency for TBE to be restricted to geographical regions provides the opportunity to focus TBE surveillance on defined risk areas. In Romania, TBE surveillance is limited to high-TBE-risk areas in the northwest of the country, comprising 11 of the 41 counties of Romania, with passive surveillance occurring from May to October [55,56]. However, the etiology of meningoencephalitis is not routinely investigated for TBEV because of low disease awareness. As such, the absence of evidence in other regions is incorrectly interpreted as evidence for the absence of TBE risk.

Incomplete surveillance can lead to a poor understanding of TBE endemic areas and potentially inadequate vaccine recommendations. This was demonstrated in Poland, where a pilot project of enhanced surveillance for TBE doubled the testing for TBE in patients with meningitis or encephalitis. As a result, 38 new endemic districts were identified, 7 of which were located far away from previously known endemic foci, highlighting the need for widespread testing for TBE in TBE-endemic countries [17].

With the observed spreading of TBE to new areas in Europe [10], restricted surveillance does not allow for the early identification of new TBE endemic areas, increasing the public’s risk of infection with TBEV. In addition, the designation of endemic or high-risk areas may limit the awareness and, thus, diagnosis of TBE in non-endemic areas, regardless of a national obligation to report TBE cases. This leads to less sensitivity in the diagnosis of imported TBE cases and the identification of TBE cases from previously non-endemic areas. In Serbia, since 2020, the Pasteur Institute Novi Sad has conducted TBE surveillance in Vojvodina province and tested all cases of viral CNS infection for the presence of anti-TBEV antibodies. Seroreactivity against TBEV was reported in 20% of tested patients who recovered from viral meningitis/encephalitis in the Vojvodina province in 2018–2019 [57]. This approach of testing all suspected TBE cases may help to identify new TBE microfoci [58] and increase awareness of travel-associated risk.

### 3.4. Active versus Passive TBE Surveillance

It is also useful to consider the type of national surveillance system employed because active TBE surveillance typically generates higher quality data than passive surveillance [59]. In the ECDC 2019 report, only the Czech Republic and Slovakia utilized active surveillance of TBE, whereby national health authorities actively seek out TBE cases [14]. Both the Czech Republic and Slovakia utilize a combination of passive and active surveillance. In the Czech Republic, active epidemiological investigation in TBE cases is mainly related to milk or cheese as a source of infection. In Slovakia, the passive component consists of the compulsory reporting of TBE laboratory results being sent daily to the Epidemiological Information System. The active component consists of daily requests to infectious disease departments about any reported TBE cases [60]. Of note, the Czech Republic performs regular surveillance of ticks in its “Prognosis of tick activity in the Czech Republic” system, whereby an interactive map highlighting areas of high tick activity is updated twice weekly for the general public [61].

TBE surveillance in the remaining countries is either mostly or entirely passive. In Croatia, TBE surveillance is entirely passive, performed via the compulsory, paper-based, infectious diseases reporting system that obliges all physicians to report all suspected infectious diseases, including TBE [62]. In Slovenia, patients with CNS inflammation are treated at infectious disease departments and all patients with aseptic meningitis/encephalitis (all having CSF pleocytosis) are tested for TBEV infection.

### 3.5. Recommendations to Improve Surveillance

Surveillance of TBE in Europe is currently incomplete, meaning that reported incidences likely only partially reflect actual risk [3,10]. Improving TBE surveillance across Europe could be achieved by:The use of a single TBE case definition across Europe to ensure that data are comparable;Testing all cases of aseptic meningitis/encephalitis of unknown etiology for TBEV infection;Rapidly expanding testing to all patients with either a fever without a known source or CNS symptoms (headache, nausea, neck stiffness, photophobia, ataxia, tremor, paralysis, polyradiculitis, and confusion) who reside in or have visited an endemic or probable or potential endemic area or have received a tick bite;Improved financing for and access to diagnostic tests and testing facilities;Employing nationwide surveillance systems in countries where these systems are lacking by implementing active surveillance systems with interactive maps of *Ixodid* tick activity throughout Europe; andIntroducing active surveillance systems across Europe.

These recommendations could be applied in an ideal healthcare setting; however, the level of implementation of these measures will largely depend on the national TBE disease burden and funding constraints. An alternative to national, active surveillance may be to employ more focused, sentinel-based TBE surveillance in TBE-endemic regions.

## 4. Current National Vaccine Recommendations

Vaccination remains the most effective method of protection against TBE, with available TBE vaccines demonstrating high seroconversion rates [19,20,21,22]. TBE vaccine recommendations vary widely across Europe (Figure 3) [10]. Currently, only Austria and Switzerland have national universal vaccination programs [10]. Other European countries with vaccine recommendations link them to certain factors, such as predefined risk areas or possible occupational exposure [63]. Vaccine recommendations in many TBE-endemic countries, such as Germany, Czech Republic, Slovenia, and Sweden, include vaccination of populations living in, working in, or traveling to TBE risk areas or endemic areas [64,65,66].

There is little clarity on how national vaccine recommendations reflect observed TBE incidence numbers; based on the completeness of current surveillance systems, the actual incidence numbers may be much higher. [10]. However, documentation of TBE-endemic areas is the minimum requirement in most countries for focused vaccination recommendations [67,68]. In Romania, the lack of TBE vaccine recommendation is in line with the lack of national recommendations for adults regarding a number of vaccines, except for the influenza vaccine. In Finland, a highly detailed map of TBE endemicity is continually updated by the National Institute of Health and Welfare, which provides highly localized vaccine recommendations to specific affected areas [69]. In Denmark, vaccine recommendations are based on areas where TBEV-infected ticks have been identified [70]. However, persons living in non-endemic areas may not be aware of the recommendation for TBE vaccination prior to visits to TBE-endemic areas.

Whereas the World Health Organization defines a highly TBE-endemic area as having an average annual pre-vaccination TBE incidence of ≥5 cases/100,000 population, to date, there is no European consensus on the definition of TBE risk areas [10,71]. The ECDC defines a risk area as one where the chances of transmission of an arthropod-borne disease to humans are higher than nil; this statement does not quantify the level of risk. An endemic area is defined as a risk area where recurrent transmission of TBE to humans is taking place over several seasonal cycles [72]. These definitions require that TBEV-infected tick populations and the number of human TBE cases be determined as accurately and completely as possible. This may not be possible in countries where surveillance is incomplete.

Beyond the abovementioned challenges in defining TBE risk areas, estimating the level of risk of TBEV infection is further complicated by annual variation in the TBEV animal reservoir; lower apparent disease incidence in areas with high vaccine uptake; and TBEV microfoci that could evade detection in otherwise TBEV-free areas [10].

Indeed, human TBE cases have been described as the tip of the iceberg, where TBEV prevalence and wildlife transmission cycles are largely hidden [73]. In the UK, TBEV was discovered in two areas with no associated confirmed cases of TBE [74]. In Poland, TBEV hotspots that had comparable proportions of infected ticks to hyperendemic regions, but with few associated TBE cases, were identified [75].

Given the difficulty in accurately identifying TBE endemic areas, the risks associated with travel to TBE-endemic areas, and the potential risk of TBEV infection in non-endemic areas, all people in TBE-endemic countries could be considered at risk of TBEV infection.

### Recommendations to Improve National Vaccine Guidance

The difficulty in identifying TBE-endemic areas combined with the unpredictable appearance of TBEV microfoci raises questions over the suitability of vaccine recommendations focused on TBE-endemic areas. As such, TBE vaccine recommendations could be expanded to encompass the whole population, rather than just those living or travelling to currently identified endemic areas. Alternatively, should this represent a significant burden on national/local healthcare services, or in countries where significant TBE risk is confined to clearly restricted areas, TBE vaccine recommendations should be harmonized so that they are easy to understand for healthcare practitioners and the public. In addition, reducing the complexity of vaccine recommendations might help the public to decode the sometimes-complex local recommendations.

## 5. Vaccine Uptake

Compounding the disparity in national vaccine recommendations are the low vaccination uptake rates across Europe. Uptake and compliance with TBE vaccination in Europe is highly variable, with low vaccination rates overall [3]. In a recent online survey of the general population in TBE-endemic European countries, only 36% of respondents had received a TBE vaccination (Figure 4). Reported vaccination rates were as high as 81% and 62% in Austria and Latvia, respectively, and as low as 7% and 11% in Romania and Poland, respectively [3,76]. Substantial regional differences in uptake were observed in Germany (data not shown) between endemic federal states (Bavaria, Baden-Wurttemberg, Thuringia, Saxony, Hesse), with an average of 60%, compared to 41% in all other federal states considered non-endemic [76].

### 5.1. Vaccine Schedule

The basic immunization protocol for TBE vaccines in Europe consists of three doses: the first vaccination dose is followed by a second dose 4–12 weeks later, and then a third dose is administered after 5–12 months [18]. An accelerated vaccine schedule is available for those traveling to TBE-endemic areas [18]. However, a recent study found that a four-dose vaccine schedule may improve antibody levels beyond 400 days in patients >50 years of age, compared with a three-dose schedule, potentially reducing the risk of vaccine failure in this age group [29].

In most of Europe, the first booster vaccination is recommended after 3 years, with subsequent booster vaccinations recommended at 5-year intervals for people <50 or <60 years of age (depending on the vaccine brand) or 3-year intervals for people >50 or >60 years of age (depending on the brand) [18]. Poor compliance to vaccine booster schedules has been observed across Europe [3]. In a European survey, whereas compliance among respondents who received ≥1 TBE vaccination was 61%, only 27% and 15% of respondents had received a first and second booster, respectively [3]. In the authors’ experience, many healthcare professionals state that the complexity of the vaccine schedules of both available TBE vaccines in Europe is a key reason why they do not vaccinate or inform their patients of the availability of the vaccine.

Indeed, in Switzerland and Finland, evidence showing that high vaccine effectiveness remained for more than 10 years following a three-dose primary series led to the adoption of a 10-year booster interval, with the hope that this may increase vaccine uptake [23,77].

### 5.2. Access to the TBE Vaccine

In some countries or regions, lack of access to the TBE vaccine may hinder vaccine uptake. TBE foci are often located in mountainous or forested areas. Rural communities may have limited access to healthcare facilities, hindering access to the TBE vaccine. In addition, in Serbia, access to the vaccine is severely limited nationwide, as permission from the Medicines and Medical Devices Agency of Serbia is required for import [78].

### 5.3. Vaccine Reimbursement

In many countries, TBE vaccination may be recommended but not reimbursed, which may reduce vaccine uptake due to economic constraints [79]. TBE vaccination is fully or partially reimbursed in only a few countries, such as Austria, Germany, and Switzerland, and is often limited to those living in endemic areas or at risk of occupational exposure [10]. In a survey of 1500 respondents in Sweden, the availability of free TBE vaccines could increase the vaccination rate by 78%, with a greater effect on low-income households [80]. The recent provision of TBE vaccination by health insurance cover in some European countries, such as the Czech Republic, could increase vaccine uptake [81].

Although many countries have a good level of vaccine uptake without a comprehensive national program [3,76], vaccine reimbursement could lead to improved vaccine uptake, particularly in low-income households [80].

### 5.4. Disease Awareness

Public awareness and knowledge of the benefits of TBE vaccination, as well as any potential consequences of non-complete immunization on vaccine effectiveness, are key to improve vaccine uptake [3]. Countries with the lowest public vaccine awareness recommend TBE vaccination in defined high-risk areas only [3]. However, vaccine awareness does not necessarily correlate with vaccine uptake, the latter being a complex, multi-factorial issue. Both high TBE awareness and TBE vaccine uptake were observed in some countries, such as Austria and Latvia, but TBE awareness was also high in some countries with low vaccination uptake rates, such as Slovakia and Romania [76].

An understanding of the risks associated with TBE may be key to improving vaccine uptake. Perception of risk or lack of risk were found to be the main drivers for vaccination or lack of vaccination, respectively, in an online survey [76]. In a survey of 1500 respondents in Sweden, trust in vaccine recommendations, perceptions about tick-bite-related health risks, and knowledge about ticks and tick-borne diseases also had significant effects on vaccination behavior [80].

Vaccine uptake can be further hindered by vaccine hesitancy, a delay in acceptance, or refusal of vaccination [82]. Vaccine hesitancy has become a major health concern in industrialized countries with fear of adverse effects, doubt around effectiveness of vaccines, and distrust toward the pharmaceutical industry stated as common reasons for non-vaccination [83]. Indeed, confidence in vaccines across Europe is low compared with other continents [84]. For TBE vaccine recommendations to be effective, the public must trust in the recommendations, understand the health risks associated with tick bites, have knowledge of TBE, and have easy access to vaccination services [80].

### 5.5. Recommendations

The complexity of the TBE vaccination schedule, accessibility, and reimbursement of the TBE vaccine, and the low awareness of potential consequences of TBE each contribute to the low uptake rates of vaccination across most of Europe [3].

Reducing the complexity of vaccination schedules may improve adherence. Investigations into the vaccination effectiveness of booster vaccines outside of the current vaccine schedules, such as that of Steffen et al. [77], are warranted to potentially support a dosing schedule beyond year 5. In addition, national vaccine reminder systems, such as those used for the COVID-19 vaccine, could be developed or improved to help improve adherence to the TBE vaccine schedule.

In countries with financial restrictions, local/targeted vaccine reimbursement programs could increase vaccination levels in certain at-risk groups. This approach has been employed in Kleszczów, Poland, where a vaccination program was introduced for people >1 year of age, targeting farmers and those spending time outdoors for recreational activity [85]. This strategy may also indirectly increase awareness of and trust in available TBE vaccines in the general population.

To improve access to vaccination, especially in rural areas, several novel approaches could be employed. Mobile vaccination initiatives, such as those employed in Finland, have proved effective in increasing local vaccine uptake and may have the additional benefit of improving disease awareness in these regions. Alternatively, drive-through vaccination centers, such as those employed successfully in several countries during the COVID-19 pandemic, may improve vaccine access in rural communities [86].

It is important to raise TBE awareness across Europe, where it is evident that the risks posed by TBE are underestimated [76]. For clinicians, in particular, primary care practitioners who are often the first contact with the patient, the provision of accurate data on the number of TBE cases and risk areas, as well as education on disease severity, is beneficial to reinforce the necessity for TBE testing.

Awareness campaigns to educate the public on the risk of TBE have already been shown to be effective. Austria has high TBE awareness and the highest TBE vaccine uptake in Europe [76]. Since 1980, Austria has implemented an annual, national TBE awareness and vaccination campaign that targets the whole population [3]. As a result, TBE cases substantially declined [79]. While broad TBE disease awareness is important, education on the potentially fatal and life-long effects of TBE, such as neurocognitive impairment, may help clinicians and patients comprehend the risk of TBE and improve risk management, including TBE vaccination uptake.

One effective approach may be to utilize clinical case studies to illustrate to patients, in a more meaningful way, the risks that TBE poses. A presentation about the existing TBE vaccines and the recommendation for vaccination could be discussed with patients that present for a consultation after a tick bite. Though vaccination will not be initiated on presentation of a tick bite, this could be an opportune moment to present the protective benefit of vaccination to patients.

## 6. Conclusions

TBE is rapidly expanding to new areas in many parts of Europe, including those at higher altitudes [10]. Although there are wide fluctuations in the reporting of national annual cases, cumulative reported TBE cases across Europe increased from 2015 through 2020, highlighting the need for better management of TBE risk in Europe [10]. TBE surveillance varies across Europe, with countries having varying diagnostic criteria, access to diagnostic tests beyond ELISAs, and approaches to national/regional surveillance [10,14,32]. Combined, these result in varying degrees of TBE underestimation across Europe [10]. Underreporting of TBE is particularly likely in children, in whom disease symptoms are often mild and may be nonspecific and, hence, may often not be correctly diagnosed [10,50,51].

Vaccination remains the most effective method of protection against TBE; however, only a few European countries have universal vaccination recommendations [10,18]. Other European recommendations link vaccination to various factors, such as predefined risk areas or possible occupational exposure, which may be inadequate given the underestimation of TBE cases and difficulty in identifying risk areas [10,63].

Vaccine uptake is low in many countries and does not appear to correlate with vaccine awareness [3,76]. While increasing vaccine awareness may improve vaccine uptake, it is also important to tackle other barriers to vaccination, such as the complexity of the vaccination schedule, the lack of TBE vaccine reimbursement, the inaccessibility of vaccination centers, and the low public perception of TBE as a severe disease.

In September and November 2021, experts in TBE from 13 European countries convened to discuss the effectiveness of European TBE surveillance systems, their impact on vaccine recommendations, and the role of vaccine awareness on vaccine uptake. The following strategies to improve TBE surveillance and vaccine uptake were recommended:Align diagnostic criteria and testing across Europe to improve the accuracy of surveillance systems and allow cross-country comparison of TBE incidence data;Improve the accuracy of surveillance systems by testing all cases of aseptic meningitis/encephalitis of unknown etiology, improving financing for and access to diagnostic tests and testing facilities, and employing nationwide surveillance systems in countries where these systems are lacking;Consider expanding vaccine recommendations to include the whole population, rather than just those living or travelling to currently identified endemic areas. Alternatively, should this represent a significant burden on national/local healthcare services, or in countries where significant TBE risk is confined to clearly restricted areas, TBE vaccine recommendations should be harmonized so that they are easy to understand for healthcare practitioners and the public;Reduce the complexity of vaccination schedules by further investigating booster vaccine effectiveness beyond year 5 to potentially increase vaccine uptake;Fund disease awareness campaigns to inform the public in TBE-endemic countries that they are at risk of TBE, a serious disease that requires vaccination with the potential for life-long complications.

A number of these recommendations may already be in place in some countries. In other countries, national or regional financial constraints may necessitate the prioritization of individual strategies over others. However, as a whole, these recommendations aim to provide a roadmap to improve TBE surveillance and vaccine uptake across Europe and reduce the TBE burden.

## Figures and Tables

**Figure 1 microorganisms-10-01283-f001:**
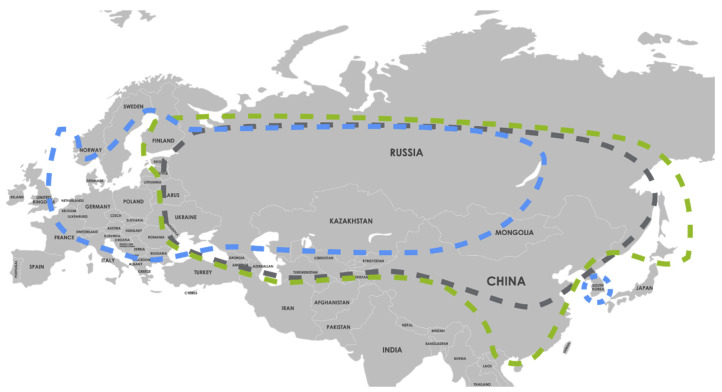
Distribution of tick-borne encephalitis virus (TBEV) subtypes in Europe and Asia. Dotted blue line, TBEV-EU: prevalent in Europe with virus isolates identified in Siberia and as far east as Lake Baikal. Dotted green line, TBEV-Sib: prevalent in Siberia and the Ural region with virus isolates identified as far west as the Baltics and Moldavia. Dotted gray line, TBEV-FE: prevalent in the far eastern region of Russia, with virus isolates identified as far west as the Baltics and Moldavia. Adapted and reprinted with permission from Ref. [10]. Copyright 2021 Global Health Press Pte Ltd.

**Figure 2 microorganisms-10-01283-f002:**
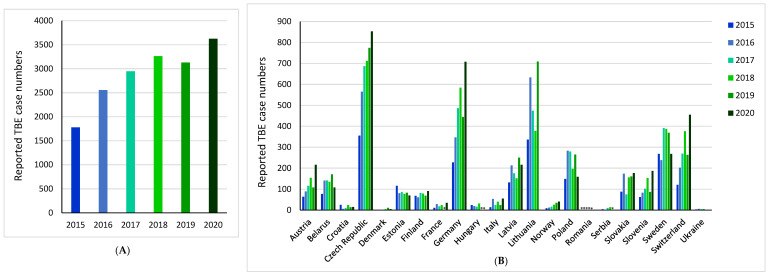
Reported tick-borne encephalitis (TBE) case numbers from 2015 to 2020: (**A**) across Europe; (**B**) per country. Totals calculated using countries with no missing reported TBE case data for the years 2015–2020. * Denotes missing data [10].

**Figure 3 microorganisms-10-01283-f003:**
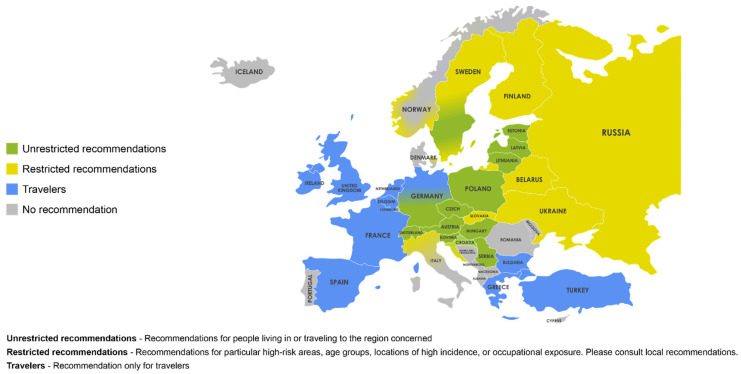
Vaccination recommendations in Europe. Unrestricted recommendation—Recommendations for people living in or traveling to the region concerned. Restricted recommendations —Recommendations for particular high-risk areas, age groups, locations of high incidence, or occupational exposure. Please consult local recommendations. Travelers—Recommendation only for travelers. Reprinted with permission from Ref. [10]. Copyright 2021 Global Health Press Pte Ltd.

**Figure 4 microorganisms-10-01283-f004:**
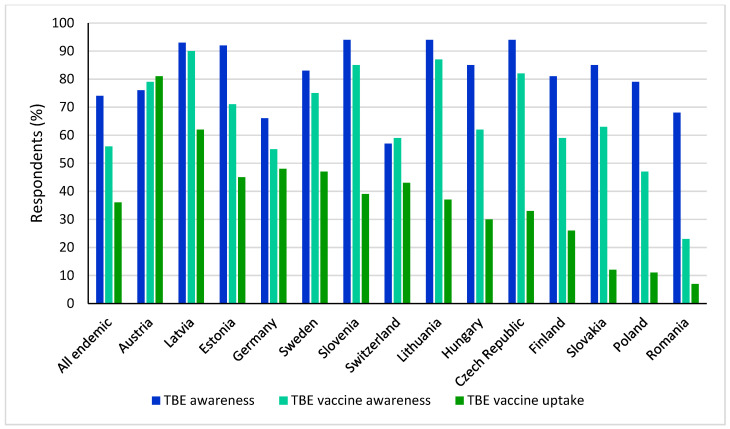
TBE disease awareness, vaccine awareness, and self-reported vaccination rate overall and across endemic countries.

## Data Availability

Not applicable.

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
