# Peer review of "Recommendations to Improve Tick-Borne Encephalitis Surveillance and Vaccine Uptake in Europe"

_microorganisms, 2022, doi:10.3390/microorganisms10071283_

Round 1

Reviewer 1 Report

TBE is neglected and there are far too many victims which could be prevented by vaccination. Thus, this is an important review, which ultimately should reduce the impact of this infection. Overall, it is well presented, but I would suggest to focus more on the essentials, on what is really news.

MAJOR COMMENTS

MINOR COMMENTS

Abstract: 

52-55 First 3 sentences — already students should know, delete.

56  "Highlighting ..." that part belongs into conclusions at the end. Suggest rather to mention that 2020 was been worst in some countries.

58  "may be" > suggest "is" — or don't you agree?

59 "TBE vaccine uptake is insufficient in much of TBE endemic ...": First, should that not rather be in 'many' than 'much'? Second, is 'insufficient' the right term? Would 'unsatisfactory', 'inadequate' not be better. Native speaker to advise, please.

Text:

69-87  Although you quote recent publications, that is essentially textbook knowledge, repeated in the introduction of many other reviews. To make the introduction attractive look for the 'news' in these articles and focus on those. 

88-92  In contrast, that must be presented with more details.

99  You should mention that this is not generally accepted and that recent studies contradict that, particularly Schmitt et al, Vaccines 2021; 9:932 — ref 23, see also the recent Zent publication in BMJ.

110  Suboptimal illustration, e.g. delete Africa and Arabian Peninsula from the map, use broader lines so that everything becomes more obvious.

124-130  Focus on Croatia — why? Suggest that you rather develop a table illustrating the increase from e.g. 2018-2019 to 2020-2021. I am aware, that these are narrow slots, possibly consider to expand the first one wider.

133  state maximum altitude as per current knowledge, please.

134(-138)  Are you really comparing single year ECDC data? In view of the annual variations associated with other factors that is a no go.

276  I doubt Gerhard Dobler would agree — what about investigating ticks?

288  Are 'numbers' the essential? Should not rather 'incidence rate' be decisive?

291  Missing word? ... except for the influenza vaccine ...

336  Be more explicit on what is meant by "complexity".

345  Uptake: Germany all country?

366  Is is NOT about seropositivity — the determining factor is field effectiveness. It is no longer just about humoral immunity, cellular immunity is the determining factor — so far neglected.

452  Suggest to add "rapidly expanding"... (not just spreading).

453  See my remark to 134, not appropriate.

481  "encompass", consider rather "include"

487-8  How?

527  How come you are referencing very old ECDC statistics — get the latest one and draw new conclusions.

Reviewer 2 Report

The review submitted for publication is a collective work of employees of 22 scientific, educational and medical institutions, dedicated to the problem of TBE, the incidence of which is widespread in the countries of the Eurasian continent and, in particular, in Western Europe. The concern of the researchers arose due to the fact that over the past 5 years (2015-2020), the incidence of TBE began to grow steadily in many countries in the study area. In this regard, in the fall of 2021, experts from 13 European countries discussed the effectiveness of European epidemic surveillance systems for the incidence of TBE and developed recommendations for the vaccine prevention of this infection.

An analysis of the literature data allowed the authors to develop and recommend common criteria for a strategy to combat the incidence of TBE for health workers in all European countries. This analysis was presented in a review article entitled: "Recommendations to improve tick-borne encephalitis surveillance and vaccine uptake in Europe". The relevance and necessity of generalized knowledge of collective work on the problem of TBE is beyond doubt, especially since the new epidemic season of 2022 has begun, which will dictate its own characteristics and dangers of this infection.

Some of the comments include:

1.      In key words: exclude flavivirus. Add words: Western Europe, incidence.

2.      Formulate the purpose of the analysis.

3.      In the text of the review, which discusses data from 21 countries (Fig. 2B), the analysis is not accurate. Authors should discuss the incidence in countries with high rates rather than repeat the analysis of reference [14].

4.      The review does not analyze which vaccines make vaccinated people sick more often.

Some recommendations have been made:

1.      In section 3.2 "Diagnostic testing", for more complete verification of TBE cases, it is possible to recommend the detection of not only IgM and IgG antibodies, but also PCR verification of a genetic marker in attached ticks, blood, and cerebrospinal fluid of the patient.

2.      In section 3.5 "Recommendations to Improve Surveillance", specify in point 6 - Implementation of an active surveillance system using interactive maps for the activity of ixodid ticks throughout Europe.

3.      Section 4 of the "Current National Vaccine Recommendations" can begin with the name of the vaccine preparations that are used in European countries for the prevention of TBE. You can show the features of these drugs so that the readers of this review can make a choice.

4.      On page 6, after line 368, a recommendation is proposed that can increase the medical and economic effect of vaccine prophylaxis against TBE. To do this, you can use the experience of other countries of the Eurasian continent, where on the eve of the next revaccination, patients are advised to check the blood for the intensity of a specific immune response. ELISA antibody values of more than 1:400 make it possible to delay the next revaccination for a year or more. (I attach a link: G.N. Leonova Influence of specific prevention on the infectious activity of the virus tick-borne encephalitis (review of own research. Russian Journal of Infectology. 2022. 14. 1: 43-52. DOI: 10.22625/2072-6732- 2022-14-1-43-52

In the conclusions, the authors made a clarification on the strategy of epidemic surveillance for TBE. The work used 85 references, where only 34% are older than 5 years, but these publications were necessary for the analysis, reflecting important aspects of the review. The review does not suffer from self-citation. The publication presents 3 figures necessary for analysis.

Round 2

Reviewer 1 Report

Much improved manuscript, thank you. Just some minor comments:

Figure 1: Make sure that the old version is deleted (it still is in the revised manuscript)

Figure 2, caption: Suggest to add the source, presumably ECDC (or just [10])?

570  Be specific, mention explicitly somewhere in the text what the current max altitude is, do not just ask the reader to consult another paper.